# Mid-Regional Proadrenomedullin in COVID-19—May It Act as a Predictor of Prolonged Cardiovascular Complications?

**DOI:** 10.3390/ijms242316821

**Published:** 2023-11-27

**Authors:** Paulina Pietraszko, Marcin Zorawski, Emilia Bielecka, Piotr Sielatycki, Edyta Zbroch

**Affiliations:** Department of Internal Medicine and Hypertension, Medical University of Bialystok, 15-089 Bialystok, Poland; pietraszko270@gmail.com (P.P.); mzorawski@wp.pl (M.Z.); emilka.bielecka96@gmail.com (E.B.); piotr.sielatycki@umb.edu.pl (P.S.)

**Keywords:** mid-regional proadrenomedullin, cardiovascular diseases, COVID-19, long-COVID-19, cardiovascular complications

## Abstract

The rising prevalence of cardiovascular disease (CVD) and the impact of the SARS-CoV-2 pandemic have both led to increased mortality rates, affecting public health and the global economy. Therefore, it is essential to find accessible, non-invasive prognostic markers capable of identifying patients at high risk. One encouraging avenue of exploration is the potential of mid-regional proadrenomedullin (MR-proADM) as a biomarker in various health conditions, especially in the context of CVD and COVID-19. MR-proADM presents the ability to predict mortality, heart failure, and adverse outcomes in CVD, offering promise for improved risk assessment and treatment strategies. On the other hand, an elevated MR-proADM level is associated with disease severity and cytokine storms in patients with COVID-19, making it a predictive indicator for intensive care unit admissions and mortality rates. Moreover, MR-proADM may have relevance in long COVID, aiding in the risk assessment, triage, and monitoring of individuals at increased risk of developing prolonged cardiac issues. Our review explores the potential of MR-proADM as a predictor of enduring cardiovascular complications following COVID-19 infection.

## 1. Introduction

Cardiovascular disease (CVD) is a group of disorders that affects the heart and blood vessels, and are among the leading causes of death and disability worldwide. The prevalence of CVD has demonstrated a substantial escalation, with incident cases witnessing a remarkable surge of 77.12% over the course of three decades, escalating from 31.31 million in 1990 to 55.45 million in 2019 [1]. CVD is connected with a common lifestyle model characterized by poor dietary choices, sedentary behavior, and tobacco use. The most common types of CVD are coronary heart disease, stroke, and heart failure. Risk factors for cardiovascular disease include hypertension, high cholesterol, diabetes, obesity, smoking, and a family history of CVD.

Preventing and managing cardiovascular disease requires a comprehensive approach including lifestyle modifications, adequate medications, and medical procedures (such as angioplasty or bypass surgery) [2].

The coronavirus disease 2019 (COVID-19) is an infectious disease caused by the SARS-CoV-2 virus that was first identified in Wuhan, China, in December 2019. COVID-19 had a significant impact on public health and the global economy. As of March 2023, there have been over 420 million confirmed cases and over 5.8 million deaths worldwide.

The spectrum of COVID-19 manifestations included fever or cough in the milder acute presentations [3], while the more severe forms were characterized by systemic complications. Obese patients, as well as those with underlying health conditions such as hypertension, diabetes, and CVD, had a higher likelihood of progressing to severe cases [4]. COVID-19 complications have the potential to persist, giving rise to long-term symptoms, as often observed in cases of “long COVID”.

The early detection and prompt treatment of cardiovascular complications both in COVID-19 patients and those with long-term effects after infection might greatly reduce negative outcomes. Therefore, it remains crucial to identify biomarkers for individuals at high risk, facilitating the improvement of prognosis and overall results.

The aim of this review is to shed light on the potential value of mid-regional proadrenomedullin as a predictor of prolonged cardiovascular complications that arise subsequent to the occurrence of a COVID-19 infection.

## 2. Characteristics of Adrenomedullin

A peptic hormone called adrenomedullin (ADM) is considered as a potential biomarker for cardiovascular disease and patients afflicted with severe infection linked to COVID-19. Adrenomedullin was named in 1993 after its initial identification in a tumor called pheochromocytoma located in the adrenal medulla [5].

An increasing body of research provides evidence that ADM plays a fundamental role in numerous biological processes. It is expressed in various tissues and organs, mainly the cardiovascular system, lungs, kidneys, and adrenal glands [6]. ADM presents a wide range of biological activities, such as anti-inflammatory and anti-apoptotic effects (Figure 1).

In vitro studies have demonstrated that various cytokines, such as tumor necrosis factor-α, -β, interleukin-1α, -β, and lipopolysaccharide, possess the capability to stimulate the release of ADM [7]. The activation of those cytokines may potentially contribute to the upregulation of ADM expression and secretion, thereby highlighting ADM involvement in immune and inflammatory responses. In addition, steroids, thyroxine, angiotensin II, noradrenaline, and bradykinin are hormones with the ability to increase ADM production.

ADM exerts its biological effects by interacting with two components: the calcitonin receptor-like receptor (CLR) and a specific receptor-activity-modifying protein. By attaching to the above receptors ADM initiates a second messenger signaling pathway, which results in an increased level of cAMP and the synthesis of nitric oxide (NO). Additionally, ADM influences various intracellular signaling pathways, including the phosphorylation of protein kinase B and the activation of protein tyrosine kinases [8].

ADM originates as a preprohormone and goes through a sequence of transformations. First, it becomes proadrenomedullin, which is subsequently fragmented into four distinct segments: proadrenomedullin N-terminal 20 peptide (PAMP)-Gly, mid-regional pro-adrenomedullin (MR-proADM), adrenomedullin-Gly, and C-terminal proadrenomedullin. Both PAMP and adrenomedullin exist as intermediate forms during this process, and they lack biological activity. The mature bioactive adrenomedullin (bio-ADM) has a brief presence in the bloodstream and tends to bind non-specifically to various surfaces, making accurate measurement challenging.

In turn, MR-proADM, comprising a 48-amino acid non-functional segment cleaved from the terminal proADM structure, reflects the ADM level and reactivity. Its prolonged half-life, spanning multiple hours, confers resilience against perturbations such as age, gender, and diurnal fluctuations. The quantification of its plasma level in clinical settings is feasible.

An enhanced MR-proADM concentration has been linked to an elevated susceptibility to mortality and adverse clinical repercussions, emphasizing its potential value in risk stratification and informed therapeutic decision making [9,10].

## 3. Relationship between Adrenomedullin and Cardiovascular Diseases

ADM operates across a spectrum of cellular components within the cardiovascular system, engaging with entities such as vascular smooth muscle cells, endothelial cells, and cardiac myocytes. ADM promotes vasodilation in the peripheral circulation, which reduces vascular resistance and lowers blood pressure [9]. Furthermore, ADM has been observed to enhance coronary flow, thereby augmenting blood supply to the coronary arteries [11]. ADM also presents cardioprotective effects, such as reducing myocardial injury and improving cardiac function in response to ischemic injury [12]. ADM has also been shown to elevate the intracellular concentration of cAMP and activate protein kinase A, thereby amplifying myocardial contractility. Additionally, ADM can exert a positive inotropic effect on myocardial cells through a cAMP-independent mechanism that involves an intracellular calcium level elevation. Nevertheless, it is important to note that the influence of ADM on contractility is heavily reliant on experimental conditions, as certain investigations utilizing human myocardial cells have failed to demonstrate a clear positive or negative inotropic effect [13].

In addition to the above, ADM has been demonstrated to trigger the protein kinase B (Akt) pathway within vascular endothelial cells to protect against myocardial injury following temporary ischemia, primarily by exerting anti-apoptotic effects. ADM also directly influences the myocardium, initiating the PI3K/Akt pathway and promoting cardioprotective effects. The combination of anti-inflammatory and anti-apoptotic effects in the cardiovascular system may contribute to a reduced risk of cardiovascular diseases, including conditions like atherosclerosis and heart failure [14].

Furthermore, ADM’s capacity to elicit an increase in renal blood flow, glomerular filtration rate (GFR), and natriuresis, all while concurrently exerting an inhibitory effect on mesangial proliferation, has been firmly established [15,16,17].

### 3.1. Adrenomedullin/MR-proADM and Hypertension

Notable observations revealed a significantly elevated plasma level of ADM among individuals diagnosed with essential hypertension in comparison to normotensive persons. This suggests a potential association between ADM and the hypertension pathophysiology. The vasodilatory impact of ADM can be attributed to various mechanisms. Firstly, ADM binds to CLR, leading to an elevation in cAMP levels within vascular smooth muscle cells. Secondly, ADM has the capacity to bind to receptors located on the endothelium, promoting NO synthesis.

In a recent study performed by Wang et al. [18], effects of ADM were particularly prominent in the context of obesity-related hypertension. ADM demonstrated the ability to improve hypertension, alleviate vascular remodeling, and reduce arterial stiffness in rats through the receptor-mediated activation of the adenosine monophosphate-activated protein kinase (AMPK) pathway.

What is more, in a study conducted by Kato et al. [19], the plasma concentration of ADM was examined in patients diagnosed with primary hypertension and malignant hypertension. The significant increase in plasma ADM concentration in subjects with malignant hypertension was identified. However, following effective hypotensive treatment, a significant decline in plasma ADM levels was observed. The heightened ADM levels in malignant hypertension could be linked to the condition severity and can be mitigated with suitable antihypertensive therapy.

Going further, in an investigation by Italian researchers [20], the study cohort was divided into subgroups based on the type of hypertension, including malignant hypertension and renovascular hypertension (the renin–angiotensin system was notably active in both groups), in comparison to individuals with primary hypertension and a normotensive control group. The results agreed with previous observations, demonstrating increased plasma ADM concentration in proportion to the hypertension severity. Significantly higher concentration of ADM was observed in malignant and renovascular hypertension, suggesting a potential contribution of the renin–angiotensin system (RAS) activation in the elevation of ADM levels.

In addition, the findings of the Jougasaki at al. study [21] indicated the presence of ADM in both renal glomerular and tubular cells. This highlights the strong natriuretic properties of ADM, suggesting its potential significance in the regulation of sodium excretion. The above observations suggest a potential role of ADM in the complex mechanisms involved in maintaining sodium balance and urinary sodium excretion.

### 3.2. Adrenomedullin/MR-proADM and Heart Failure

Heart failure (HF) is a multifaceted medical condition that arises when the heart’s ability to pump an adequate amount of blood throughout the body becomes inadequate.

ADM’s enhancement of cardiac output can be explained by several factors, including a systemic vascular resistance reduction and coronary blood vessel expansion. ADM also plays a role in elevating cAMP levels and activating protein kinase A, thereby strengthening the contractile force of the heart muscle. Furthermore, ADM can improve the contractility of heart cells through a mechanism independent of cAMP, causing an elevation in intracellular calcium levels.

The concentration of plasma ADM shows a positive correlation with the degree of HF severity [22]. A study conducted by Yu [23] found that patients diagnosed with systolic and isolated diastolic heart failure exhibited higher levels of plasma adrenomedullin, with a notable elevation observed in cases where a restrictive filling pattern was present.

Based on the results of the Biomarkers in Acute Heart Failure study [24], MR-proADM demonstrated greater precision in predicting all-cause mortality within 90 days among patients admitted with acute heart failure, surpassing the accuracy of B-type natriuretic peptide (BNP). Notably, MR-proADM presented as the primary contributor to the overall predictive performance observed. The validity of this observation was reinforced through a sub-analysis conducted within a different study [25]. The results revealed that MR-proADM emerged as the most effective indicator of one-year mortality in cases of acute heart failure. However, both MR-proADM and N-terminal pro-B-type natriuretic peptide (NT-proBNP) demonstrated a superior prognostic value beyond the initial year following the diagnosis of acute heart failure.

In a recent study [26], the researchers assessed the influence of MR-proADM on the two-year survival rate of patients with chronic heart failure, while establishing its association with the dosage of furosemide. Significant statistical disparities (*p* < 0.001) were observed across multiple parameters: individuals with elevated MR-proADM levels displayed advanced age, higher New York Heart Association (NYHA) classification, an increased incidence of lower limb edema, and a higher prevalence of comorbidities including hypertension, atrial fibrillation, diabetes, and renal impairment. Moreover, there was a correlation between MR-proADM levels and the dosage of furosemide. Patients receiving higher diuretic doses presented elevated MR-proADM levels. The findings suggested that a higher plasma concentration of MR-proADM in individuals diagnosed with chronic heart failure was connected with a heightened susceptibility to mortality and hospitalization. Moreover, elevated MR-proADM levels in conjunction with an intensified usage of loop diuretics indicated the persistence of congestion and was connected with a higher likelihood of severe disease advancement.

Another research study was conducted to assess the diagnostic value of MR-proADM as a biomarker in screening tests for a left ventricular hypertrophy (LVH) in patients with hypertension [27]. LVH is a compensatory mechanism that occurs in response to left ventricular pressure overload and is among the potential hypertension complications. The study revealed that patients with hypertension and LVH exhibited a significantly higher MR-proADM plasma concentration compared to those without LVH, as determined by echocardiography. Furthermore, according to the authors’ suggestions, measuring MR-proADM plasma levels could be practically useful as a rule-out test for LVH in hypertensive patients. Nevertheless, given the constrained specificity of the test, it is advisable that patients who yield a positive screening result, indicating the presence of an optimal cutoff value for MR-proADM as an indicator of LVH, undergo subsequent evaluation through echocardiography for further assessment and confirmation.

In a contemporary inquiry concerning acute heart failure [28], a notable discovery surfaced, indicating that MR-proADM exhibited diagnostic accuracy akin to that of the established gold-standard biomarker NT-proBNP within acute heart failure (AHF) cohorts. Furthermore, the conjoined implementation of MR-proADM and NT-proBNP, both upon admission and discharge, yielded a substantial augmentation in prognostic and predictive capacities both for mortality rates and the incidence of renewed hospitalization due to AHF within the limited temporal span of 30 days post-discharge. While NT-proBNP traditionally encapsulates myocardial responses to hemodynamic stress and functional overburden, the inclusion of MR-proADM provides distinctive insights into the quantum of oxidative stress as well as the overall disease severity. The collective integration of these two biomarkers facilitates a more comprehensive and nuanced assessment of the underlying pathological mechanisms encompassing the realm of acute heart failure.

### 3.3. ADM/MR-proADM and Myocardial Infarction

ADM participates in the modulation of several biological processes, including vasodilation, the suppression of platelet aggregation, and the induction of anti-inflammatory effects, during myocardial infarction (MI). One of the initial research articles focused on investigating the concentration of ADM in patients with MI [29]. The authors measured ADM levels in two different moments: one day after the myocardial infarction occurrence and again four weeks later. They observed that the plasma ADM concentration was elevated during the early stages of acute myocardial infarction, with the magnitude of elevation directly corresponding to the clinical severity of the condition. What is more interesting, patients diagnosed with congestive heart failure demonstrated even an higher concentration of plasma ADM.

Moreover, the LAMP (Leicester Acute Myocardial Infarction Peptide) Study [30] revealed a notable rise in plasma MR-proADM level following myocardial infarction, demonstrating a strong association with unfavorable cardiac outcomes. Furthermore, Arrigo et al. [31] reported a positive correlation between the concentration of MR-proADM and the severity of acute MI, with the highest concentration observed in patients experiencing cardiogenic shock. The study also revealed an association of elevated MR-proADM levels at the time of admission with an increased risk of acute HF during hospitalization.

The described findings are consistently supported by current research, as exemplified by a 2022 investigation [32] that concentrated on enrolling individuals who had experienced an acute phase of ST-segment elevation myocardial infarction (STEMI). The participants, who were admitted to a cardiac intensive care unit, were followed up for a period of 90 days after discharge from hospitalization. The study revealed a significant increase of the MR-proADM concentration during the acute phase of STEMI, which successively declined during the recovery phase. Importantly, higher levels of MR-proADM during the acute phase of STEMI, as well as the intensity of its subsequent change, were identified as predictive factors for adverse cardiac events occurring within the 90-day follow-up period.

Another interesting research study by Tzikas et al. [33] included individuals who were admitted with an acute chest pain suggestive of an acute coronary syndrome. Notably, a higher concentration of MR-proADM was observed in patients who experienced either death or a non-fatal myocardial infarction. Additionally, the incorporation of MR-proADM into the Global Registry of Acute Coronary Events (GRACE) risk score led to a substantial reclassification of patients, resulting in an overall net reclassification improvement of 41.2% for MR-proADM. The study concluded that MR-proADM can serve as a valuable predictor of future cardiovascular events in individuals who present with acute chest pain.

### 3.4. Adrenomedullin as a Potential Therapeutic Agent in Cardiovascular Diseases

Adrenomedullin has been studied as a potential therapeutic agent for cardiovascular diseases due to its multiple physiological effects, including vasodilation and the regulation of blood pressure (Table 1).

Nagaya et al. [39] explored the potential application of ADM for the treatment of patients diagnosed with acute myocardial infarction. Based on the findings, it can be stated that the intravenous administration of adrenomedullin resulted in a notable enhancement of left ventricular myocardial contraction and improved left ventricular relaxation in patients following MI. Notably, the beneficial effect was observed without a concurrent increase in myocardial oxygen consumption (MVo2). In contrast to another potential biomarker, atrial natriuretic peptide (ANP), ADM demonstrated a distinct profile of myocardial effects, suggesting its potential as a valuable therapeutic intervention in the post-myocardial infarction setting.

Another study revealed that the administration of ADM through venous infusion resulted in the dilation of pulmonary arteries and the reduction of pulmonary arterial pressure [40]. However, it is important to note that this intervention also led to a decrease in systemic blood pressure. Despite the above-mentioned likely benefits, the challenges associated with the handling of ADM, along with its inability to surpass the efficacy of currently available medications for pulmonary hypertension, contribute to its current unsatisfactory status as a treatment option for this particular condition.

While the data presented here provide evidence supporting the value of MR-proADM as a prognostic indicator, especially in the context of short-term risk stratification, certain limitations hinder its application in clinical practice. Worth noting is the observation that MR-proADM is not exclusively specific to cardiac conditions, as its expression was observed in various anatomical sites within the body. Additionally, a comprehensive characterization of factors that might impact the interpretation of MR-proADM level is still underway. The translation of experimental findings into clinical practice through rigorous clinical trials is essential to establish safety, efficacy, and potential therapeutic applications in the management of cardiovascular conditions.

## 4. Significance of ADM and MR-proADM in COVID-19

Endothelial dysfunction caused by the virus is considered to be a significant contributor to the varied symptomatology observed in COVID-19 [41]. This deterioration results in compromised hemodynamics, augmented coagulation tendencies, enhanced capillary permeability, and the manifestation of edema. Additionally, even post-resolution of COVID-19, lingering endothelium-dependent dysfunction persists [42,43]. Prolonged inflammation within cardiac tissue can progress, possibly resulting in various manifestations, such as a reduced flexibility of the ventricles due to increased stiffness and compromised contractile function as a result of inadequate blood supply. Furthermore, persistent inflammation, fibrosis, and a lasting reduction in blood supply promote arrhythmias [44,45].

Previous studies demonstrated that both ADM and MR-proADM levels increased during inflammatory disorders as a means to modulate microcirculation and counteract endothelial hyperpermeability. Furthermore, the measurement of ADM and MR-proADM plasma concentration was considered as a valuable indicator for assessing the severity of endothelial damage [10,46]. MR-proADM plays a pivotal role as a marker in inflammation, vascular permeability, and microcirculation stability. These multifaceted processes collectively contribute to the development of organ dysfunction and failure in various pathological conditions [47,48]. Therefore, MR-proADM also fulfills a significant role in the identification and prognosis of sepsis and septic shock.

### 4.1. Impact of MR-proADM on Mortality in COVID-19

The rapid spread and extensive repercussions of the COVID-19 pandemic have accelerated comprehensive science investigations, particularly evident in the study of sepsis, where a substantial amount of research has concentrated on understanding the role of adrenomedullin. ADM reduces tissue damage by enhancing the regulation of cytokines and fortifying the barrier function of endothelial cells. Thereby, the link between elevated levels of ADM and the acuteness of the disease’s clinical presentation has been established [49,50,51].

In a study conducted by Johnsson et al. [52], the findings indicated that an elevated bio-ADM level detected upon admission was associated with the development of acute respiratory distress syndrome (ARDS), regardless of the presence of sepsis and organ dysfunction, as evaluated by the Sequential Organ Failure Assessment (SOFA) score. Importantly, both a lower (below 38 pg/L) and a higher (above 90 pg/L) bio-ADM concentration independently served as a prognostic biomarker for mortality, irrespective of the Simplified Acute Physiology Score (SAPS-3). Furthermore, patients with indirect mechanisms of lung injury demonstrated higher bio-ADM levels in comparison to those with a direct injury mechanism. What is even more interesting, the bio-ADM concentration tend to rise with the severity of ARDS.

Wang et al. [53], in a comprehensive meta-analysis, compared MR-proADM concentrations between two groups of COVID-19 patients, survivors and non-survivors, with a primary emphasis on mortality as the endpoint. The study confirmed a notable disparity in MR-proADM levels between the two patient groups. Elevated MR-proADM levels were independently associated with mortality in COVID-19 patients, affirming its effectiveness in assessing the risk. These results underscored the substantial predictive capability of MR-proADM in identifying adverse outcomes among COVID-19 patients.

Van Lier et al. [54] investigated the 28-day mortality rates among COVID-19 patients admitted to the intensive care unit. Their research study revealed a significant link between bio-ADM levels and short-term mortality, as well as the development of acute kidney injury in critically ill COVID-19 patients in various stages. The described findings align with previous studies, emphasizing the potential importance of bio-ADM as a prognostic marker for COVID-19 patients.

### 4.2. Comparison of MR-proADM with Other Biomarkers and Critical Illness Scores in Outcome Prediction

To date, several research efforts have been dedicated to elucidating the function and significance of MR-proADM in comparison to other biomarkers and critical illness scores. Of note, an interesting clinical study suggested that decisions guided by risk scores showed lower effectiveness when compared to those grounded in biomarker analysis [55].

One particular investigation demonstrated that the prognostic value of MR-proADM for predicting mortality surpassed the SOFA score, Acute Physiological and Chronic Health Evaluation IV (APACHE IV), Confusion blood Urea nitrogen Respiratory rate Blood pressure age 65 or older (CURB-65) score, D-dimer, cardiac troponin T (cTNT), C-reactive protein (CRP), procalcitonin (PCT), ferritin, lactate, and C-terminal proendothelin-1 (CT-proET-1) [56].

Furthermore, the researchers observed that the assessment of MR-proADM was superior to PCT, creatinine, albumin, platelet count, interleukin-6 (IL-6), and lymphocyte count, as demonstrated by the area under the curve (AUC) [57].

In turn, Benedetti et al. [58] demonstrated MR-proADM’s superior predictive ability compared to the SOFA score and the Simplified Acute Physiology Score II (SAPS II), although it was relatively weaker when compared to the Acute Physiology and Chronic Health Evaluation II (APACHE II) score.

A recent prospective observational study involving a cohort of over 200 critically ill patients with COVID-19 in the ICU further indicated the prognostic value of MR-proADM in mortality predicting. The researchers examined several biomarkers including D-dimer, LDH, NT-proBNP, PCT, myoglobin, troponin-I (hs), neutrophils, lymphocytes, natural killer lymphocytes, and IL-6, all measured within the initial 48 h of ICU admission. Significantly different levels of those biomarkers were observed between patients who survived and those who did not. Of particular interest, the MR-proADM plasma concentration measured within the same 48 h window was notably higher in non-surviving patients. In fact, when comparing the predictive abilities of different biomarkers, MR-proADM presented the highest level of accuracy. Even when combining multiple biomarkers, there was no observed improvement in predictive capability beyond that achieved by MR-proADM alone. What is more, MR-proADM values exceeding 1.5 nmol/L, indicating a higher predictive value, exhibited elevated mortality rates, thus substantiating the potential usefulness of that threshold as a cutoff value within this particular population [59].

In summary, there has been a concurrent assessment of the relationship between MR-proADM and various other biomarkers in the context of COVID-19. The findings consistently demonstrated the superiority of MR-proADM over other biomarkers [60,61]. Despite the well-established significance of MR-proADM in outcome prediction, its integration into routine clinical practice necessitates further evaluation through large-scale trials.

### 4.3. ADM as a Therapeutic Agent in COVID-19

Over the past few years, numerous investigations have aimed to explore the impact of administering ADM, modifying its function, or blocking its effects on outcomes in preclinical models of sepsis, systemic inflammation, and organ injury [62,63,64,65,66]. The administration of ADM has shown promising results in improving hemodynamics, reducing vascular permeability, mitigating organ damage, and enhancing outcomes in different experimental models of endotoxemia.

The confirmed correlation between adrenomedullin and COVID-19 underscores the importance of conducting clinical trials to explore the potential use of ADM as a therapeutic intervention for treating the disease. By verifying the beneficial effects of ADM, it could emerge as an innovative treatment option for severe COVID-19 or other infection, offering significant advantages for public health.

Two ongoing phase 2a clinical trials, employing a randomized, double-blind, multicenter, and placebo-controlled design, are currently underway. Valuable findings are expected and the results are anticipated to be disclosed in the near future [67].

## 5. Knowledge Gap in MR-proADM Levels and Cardiovascular Risk in COVID-19—Potential Implications for Long COVID

Cardiovascular complications are frequently observed among patients diagnosed with COVID-19 [68]. Previous research has established a clear association between ADM and cardiovascular diseases, as well as between ADM and COVID-19. However, there is currently a dearth of data regarding the correlation between MR-proADM concentration in COVID-19 patients and the risk of cardiovascular complications. Given this knowledge gap, there exists a requirement to investigate whether MR-proADM could serve as a valuable biomarker for identifying COVID-19 patients who are at an elevated risk of developing cardiovascular complications.

Interestingly, the concept of “long COVID” has recently gained prominence, describing a range of symptoms that persist well beyond the acute phase of the illness. Cardiovascular complications, including myocardial inflammation and impaired cardiac function, have been reported among individuals experiencing long COVID [69]. Alterations in cardiovascular conditions have additionally been discerned using imaging techniques [70].

Recent investigations have provided compelling evidence that individuals with a prior history of COVID-19 are at a substantially elevated risk of experiencing new cardiovascular events following the acute phase, as compared to those who have not been afflicted by the virus. The array of complications observed in prolonged cases of COVID-19 encompasses a wide spectrum of cardiac issues, ranging from left and right ventricular systolic dysfunction [71,72,73,74,75,76], arrhythmias [77], pulmonary hypertension [73,78,79], pericarditis [73,80], and heart failure [81,82], to more acute conditions such as acute myocardial infarction [82,83], coronary artery disease [84], the formation of left ventricular thrombi [84], and myocarditis [76,82,85].

Consequently, maintaining ongoing surveillance for the manifestation of cardiovascular conditions beyond the initial 30-day timeframe subsequent to a COVID-19 diagnosis is of paramount importance [86,87]. This phenomenon presents a new avenue for investigating the potential effects of both ADM and MR-proADM in post-acute COVID-19 patients.

Recent studies have suggested that long COVID may have an autoimmune component, with continual inflammation contributing to various symptoms [88,89,90,91,92]. Given ADM’s anti-inflammatory properties and its role in mitigating cardiovascular damage, it is plausible that altered ADM levels might influence the risk of cardiovascular complications in long COVID patients. Investigating the correlation between ADM concentration and cardiovascular outcomes in the context of long COVID could provide valuable insights into disease mechanisms and potential therapeutic targets.

## 6. Conclusions

The growing elderly population, as well as the increasing prevalence of cardiovascular disease, and the ongoing global health crisis due to SARS-CoV-2 infection emphasized the need for searching for new biomarkers to assess the progression and prognosis of COVID-19 and other infective diseases. The existing body of knowledge underscores the critical role of ADM in cardiovascular health and its potential implications in COVID-19. However, the lack of comprehensive data regarding the association between MR-proADM levels in COVID-19 patients and the risk of developing cardiovascular complications highlights a significant gap in current research. Exploring this link could potentially lead to the identification of MR-proADM as a valuable biomarker for predicting cardiovascular risk in COVID-19 patients and individuals experiencing long COVID, offering opportunities for early intervention and improved patient outcomes.

## Figures and Tables

**Figure 1 ijms-24-16821-f001:**
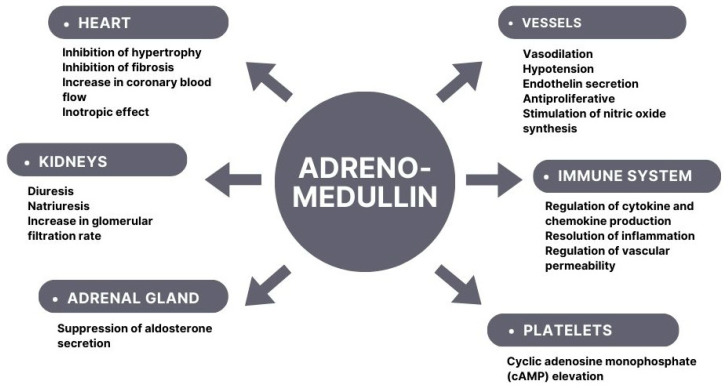
Adrenomedullin influence on organ and system functions.

**Table 1 ijms-24-16821-t001:** Studies evaluating the effects of ADM on heart failure.

Author	Population	Effect
Nakamura (1997) [34]	Humans	vasodilatation, impaired response in patients with congestive heart failure
Szokodi (1998) [35]	Rats	augmentation of cardiac contractility
Nagaya (1999) [11]	Rats	increase in urine production and sodium excretion without causing hypotension, dilation of renal blood vessels
Saetrum Opgaard (2000) [12]	Humans	no direct effects on myocardial contractility
Jougasaki (2001) [36]	Dogs	attenuation of renal natriuretic responses to ADM in experimental acute heart failure
Rademaker (2002) [37]	Sheep	decrease of cardiac preload and afterload, increased cardiac output
Nishikimi (2009) [38]	Humans	combination of ADM and human atrial natriuretic peptide (hANP): decreased resistance in systemic and pulmonary blood vessels, increased cardiac output

## Data Availability

No new data were created or analyzed in this study. Data sharing is not applicable to this article.

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
