# Peer review of "Mid-Regional Proadrenomedullin in COVID-19—May It Act as a Predictor of Prolonged Cardiovascular Complications?"

_ijms, 2023, doi:10.3390/ijms242316821_

Round 1

Reviewer 1 Report

Comments and Suggestions for Authors

The title of this manuscript is very attractive, but there are no data to support this concept directly. Entire composition of the paper is mediocre and presented data have seen before in other papers. And thus this review don’t have enough impact for the readers. Other concerns are following.

1.      The description of pulmonary hypertension, line 128 to 140, is unnecessary for this hypertension segment. In addition, the content is not worth for entire manuscript. MR-proADM is not the precursor of ADM, it is non-functioning fragment of the precursor, line 128.

2.      Line 310 and 313, lethal confusion of ADM and MR-proADM was occurred. MR-proADM is non-functioning fragment and do not have any bioactivity. Line 315 to 323 is also unnecessary for COVID-19 explanation.

3.      Line 445, MR-proADM may be ADM.

4.      The list of references is very confused and should be unified to standard style. For example, No 19 contained two articles with different style. No 17 and others lost journal name and following information.

Author Response

We want to express our gratitude for the thorough review of our manuscript.
We appreciate the time and efforts in reviewing this manuscript. We have addressed all issues indicated in the review reports and believed that the revised version can meet the journal publication requirements.

Reply:

Comment: The title of this manuscript is very attractive, but there are no data to support this concept directly. Entire composition of the paper is mediocre and presented data have seen before in other papers. And thus this review don’t have enough impact for the readers. Other concerns are following.

            We appreciate your valuable comment. We’d like to explain that our paper did not aim to introduce entirely new data. Instead, we focused on consolidating and examining existing data and research to provide a comprehensive overview of the subject matter. Additionally, we aimed to present the concept of potential involvement and impact of mid-regional proadrenomedullin in the context of COVID-19 as a potential predictor of prolonged cardiovascular complications, what wasn’t extensively addressed in the previous literature. We believe that this novel approach can provide valuable insights for readers, as it highlights the need for further investigations and potential avenues for future research.

1 The description of pulmonary hypertension, line 128 to 140, is unnecessary for this hypertension segment. In addition, the content is not worth for entire manuscript. MR-proADM is not the precursor of ADM, it is non-functioning fragment of the precursor, line 128.

            The description of pulmonary hypertension, including inaccurate description of MR-proADM was deleted.

2 Line 310 and 313, lethal confusion of ADM and MR-proADM was occurred. MR-proADM is non-functioning fragment and do not have any bioactivity. Line 315 to 323 is also unnecessary for COVID-19 explanation.

            We corrected the text in lines 310-313, so it would not suggest that MR-proADM is bioactive. Furthermore, the text in lines 315-323 was included into the paragraph 3.3.

3 Line 445, MR-proADM may be ADM.

            Both - ADM and MR-proADM were taken into consideration.

4 The list of references is very confused and should be unified to standard style. For example, No 19 contained two articles with different style. No 17 and others lost journal name and following information.

 We did the suggested corrections.

Reviewer 2 Report

Comments and Suggestions for Authors

In this manuscript (ID# ijms-2664272), entitled “Mid-regional proadrenomedullin in COVID-19 - may it act as a predictor of prolong cardiovascular complications?”, authors Pietraszko et al reviewed the information regarding MR-proADM as a possible biomarker in cardiovascular damage in COVID-19 viral infection. The association between ADM elevation and other cardiovascular diseases is also provided, making this manuscript more interesting. There are several concerns, which are listed in the following paragraphs:

1. Please define the terms: ADM, proADM, bio-ADM, and MR-proADM in the introduction. In the manuscript, the different names are used, which is confusing.  What is their relationship? How they are synthesized and degraded in the cells or plasma. Which one is the best predictor for cardiovascular damage? Why? This information is the basic background knowledge to understand their biofunction in the diseases involved.

2. What is the possible mechanism involved in the elevated ADM levels in those cardiovascular diseases?

3. Table 2 is unnecessary. It would be better to described in the text, instead of a table. 

Comments on the Quality of English Language

The English Language is fine, just needs minor edition.

Author Response

We want to express our gratitude for the thorough review of our manuscript.
We appreciate the time and efforts in reviewing this manuscript. We have addressed all issues indicated in the review reports and believed that the revised version can meet the journal publication requirements.

Reply:

1 Please define the terms: ADM, proADM, bio-ADM, and MR-proADM in the introduction. In the manuscript, the different names are used, which is confusing.  What is their relationship? How they are synthesized and degraded in the cells or plasma. Which one is the best predictor for cardiovascular damage? Why? This information is the basic background knowledge to understand their biofunction in the diseases involved.

            A more in-depth description was added to explain the terms and their relationship.

  1. What is the possible mechanism involved in the elevated ADM levels in those cardiovascular diseases?

            The information was added in the paragraph 3.

  1. Table 2 is unnecessary. It would be better to described in the text, instead of a table. 

            We did the suggested corrections.

Round 2

Reviewer 2 Report

Comments and Suggestions for Authors

No further recommendation